# Matrix Metalloproteinase Genes (*MMP1, MMP10, MMP12*) on Chromosome 11q22 and the Risk of Non-Contact Anterior Cruciate Ligament Ruptures

**DOI:** 10.3390/genes11070766

**Published:** 2020-07-08

**Authors:** Ewelina Lulińska, Andrea Gibbon, Mariusz Kaczmarczyk, Agnieszka Maciejewska-Skrendo, Krzysztof Ficek, Agata Leońska-Duniec, Michał Wilk, Katarzyna Leźnicka, Monika Michałowska-Sawczyn, Kinga Humińska-Lisowska, Rafał Buryta, Paweł Cięszczyk, Ewelina Maculewicz, Wojciech Czarny, Alison V. September, Marek Sawczuk

**Affiliations:** 1Faculty of Physical Culture, Gdansk University of Physical Education and Sport, 80-336 Gdansk, Poland; wzspdyrektor@gmail.com (E.L.); mariush@pum.edu.pl (M.K.); maciejewska.us@wp.pl (A.M.-S.); leonska.duniec@gmail.com (A.L.-D.); k.leznicka@tlen.pl (K.L.); monikamichalowska@op.pl (M.M.-S.); kinga.huminska@gmail.com (K.H.-L.); cieszczyk@poczta.onet.pl (P.C.); sawczuk_marek@wp.pl (M.S.); 2Division of Exercise Science and Sports Medicine, Department of Human Biology, University of Cape Town, PO Box 115, Newlands 7725, Cape Town, South Africa; loggiboggi01@gmail.com (A.G.); alison.september@uct.ac.za (A.V.S.); 3Instutute of Physical Culture and Health Promotion Sciences, University of Szczecin, 70-453 Szczecin, Poland; rafalburyta@poczta.onet.pl; 4Institute of Sport Sciences, The Jerzy Kukuczka Academy of Physical Education in Katowice, 40-065 Katowice, Poland; krzysztof.ficek@galen.pl; 5Galen-Orthopaedics, 43-150 Bierun, Poland; 6Department of Biomedical Sciences, Faculty of Physical Education, Jozef Pilsudski University of Physical Education in Warsaw, 00-809 Warsaw, Poland; ewelina.jask@gmail.com; 7College of Medical Sciences, Institute of Physical Culture Studies University of Rzeszów, 35-959 Rzeszów, Poland; wojciechczarny@wp.pl

**Keywords:** metalloproteinase, MMP, gene polymorphism, ACL rupture

## Abstract

Background: Sequence variants within the matrix metalloproteinases genes remain plausible biological candidates for further investigation of anterior cruciate ligament (ACL) rupture risk. The aim of the present study was to establish whether variants within the *MMP1* (rs1799750, ->G), *MMP10* (rs486055, C > T) and *MMP12* (rs2276109, T > C) genes were associated with non-contact ACL rupture in a Polish cohort. Methods: The unrelated, self-reported Polish Caucasian participants consisted of 228 (157 male) individuals with primary non-contact ACL rupture and 202 (117 male) participants without any history of ACL rupture. All samples were genotyped in duplicate using the Applied Biosystems TaqMan^®^ methodology. The statistical analyses were involved in determining the distribution of genotype and allele frequencies for the investigated polymorphisms between the diagnostic groups. Furthermore, pseudo-haplotypes were constructed to assess possible gene–gene interactions. Results: All genotype frequencies in the ACL rupture and control groups conformed to Hardy Weinberg Equilibrium expectations. None of the polymorphisms were associated with risk of non-contact ACL rupture under the codominant, dominant, recessive and over-dominant genetic models. Likewise, no genotype–genotype combinations inferred as “haplotypes” as a proxy of gene–gene interactions were associated with the risk of non-contact ACL ruptures. Conclusions: Despite the fact that the current study did not support existing evidence suggesting that variants within the *MMP1*, *MMP10*, and *MMP12* genes influence non-contact ACL rupture risk, future work should include high-throughput sequencing technologies to identify potential targeted polymorphisms to fully characterize the 11q22 region with susceptibility to non-contact ACL rupture susceptibility in a Polish cohort.

## 1. Introduction

The anterior cruciate ligament (ACL) is one of the most commonly injured ligaments of the knee, therefore, more research is needed to increase our understanding of the etiology, mechanisms and risk factors for ACL injury [1]. A number of risk factors affect ACL injuries, both internal (intrinsic) as well as external (extrinsic). Intrinsic factors include neuromuscular and cognitive function, hormonal milieu, anatomic variables, sex, as well as genetic factors [2,3]. Based on strong scientific evidence, it is suggested that due to variations in certain genes or DNA sequences some individuals may be more prone to ACL injuries than others [4,5,6].

The fibrous connective tissue of tendons and ligaments is composed of numerous collagenous fibre types (e.g., I, III–VI and, XII) and minor elastic fibres (e.g., elastin) as well as other non-collagenous particles such as proteoglycans (e.g., decorin, lumican and aggrecan) and glycoproteins (e.g., tenascin C) [7]. Matrix metalloproteinases (MMPs) are a large family of more than 25 structurally related, zinc- and calcium-dependent endopeptidases that function to maintain baseline extracellular matrix (ECM) homeostasis, through modulating the structural and biological integrity of the tendons and ligaments [8,9]. Since MMPs play a crucial role in the remodeling of ECM, the dysregulation of their activity may lead to alterations in normal ECM architecture and disruptions in normal force transmission which may consequently influence injury or disease development. What is more, functional polymorphisms in the MMP genes may alter gene and/or protein activity, which could disrupt the ECM balance and thus may result in excessive ECM destruction [10].

It has been proposed that MMP genes’ sequence variants may play a role in the aetiology of both exercise- and occupational-associated acute and chronic musculoskeletal soft tissue injuries. Variants within *MMPs* have previously been associated with numerous complex musculoskeletal disorders such as rheumatoid arthritis [11], osteoarthritis [12], lumber disk degeneration [13] and idiopathic scoliosis [14]. Furthermore, several genetic case-control association studies have identified sequence variants within the *MMP1* (rs1799750), *MMP3* (rs679620, rs591058, rs650108, rs3025058), *MMP10* (rs486055) and *MMP12* (rs2276109) genes, which are all located on chromosome 11q22 [15], to be associated with Achilles tendinopathy (AT) [16,17,18] and anterior cruciate ligament (ACL) rupture [19,20].

Recently, four single nucleotide polymorphisms (SNPs) located within the *MMP3* (rs591058, rs679620), *MMP8* (rs11225395), and tissue inhibitor of metalloprotease 2 (*TIMP2*, rs4789932) genes were investigated as a possible risk factors for non-contact ACL ruptures in a Polish population [21]. The results from this study demonstrated the significant association of the *MMP3* rs679620 and rs591058 polymorphisms with non-contact ACL rupture risk. These findings further support the previous work by Posthumus et al. [20], suggesting that genetic variation within *MMP3* may contribute to the inter-individual susceptibility to non-contact ACL ruptures.

Based on the collective findings thus far, the genes encoding MMPs, particularly those spanning the chromosome 11q22 region, remain plausible biological candidates for further investigation into the genetic basis of ACL rupture risk. Therefore, in order to further define the predisposing genetic profile mapping to the 11q22 chromosomal region, the present study aimed to establish whether variants within the collegenase 1 (*MMP1*, rs1799750 - >G), stromelysin 2 (*MMP10*, rs486055 C > T) and macrophage metalloelastease (*MMP12*, rs2276109 T > C) genes were associated with non-contact ACL rupture in a Polish cohort. Secondly, this study aimed to determine whether pseudo-haplotypes constructed from the investigated variants within the *MMP1*, *MMP10* and *MMP12* genes modulated the risk of non-contact ACL ruptures.

## 2. Materials and Methods

A total of 430 physically active, unrelated, self-reported Caucasian participants were recruited for this case-control genetic association study between the years 2009 and 2016. The participants consisted of 228 (157 male) individuals with surgically diagnosed primary ACL rupture who qualified for ligament reconstruction (ACLR group) and 202 (117 male) apparently healthy control participants, all of whom reported no previous ACL injuries (CON group). All 228 participants from the ACLR group sustained their injury through non-contact mechanisms [22]. The ACLR participants were soccer players (157 males and 71 females) from the Polish 1st, 2nd and 3rd division soccer league (trained 11–14 h per week; mean time: 11.9 ± 1.4 h). The control group consisted of apparently healthy, physically active individuals, the majority of whom reported soccer as their primary sport, with no previous ligament and/or tendon injuries. All the male participants were ancestrally matched (all self-reported Polish, East-Europeans for ≥3 generations), of a similar age (ACLR group: =26 ± 4 years, CON group: =26 ± 6 years), and had comparable levels of sporting exposure (same volume and intensity of training and match play). The ACLR female participants (mean age: 26 ± 6 years) were soccer players from the Polish 1st and 2nd division soccer league (trained 10–12 h per week; mean time: 11.1 ± 0.6 h). The female control participants (mean age: 29 ± 2 years) were recruited from sports clubs and wellness centers and self-reported as being physically active for a minimum of 7 h per week (mean time: 9.2 ± 1.4 h).

The procedures followed in the study were conducted in accordance with the principles of the World Medical Association Declaration of Helsinki and were approved by the Ethics Committee of the Pomeranian Medical University in Szczecin (approval numbers BN-001145/08 and KB-0012/102/10), Ethics Committee at the Regional Medical Chamber in Szczecin (approval numbers 09/KB/IV/2011 and 01/KBNI/2017) and Ethics Committee at the Regional Medical Chamber in Gdańsk (approval number KB-8/16). All participants were provided with information sheet concerning study particulars including, the purpose of the study and the procedures involved, in addition to the possible risks and benefits associated with participation. All participants provided written informed consent to genotyping on the understanding that it was anonymous and that the obtained results would be confidential. This case-control genetic association study was conducted in accordance with the set of guiding principles for reporting the results of genetic association studies defined by the Strengthening the Reporting of Genetic Association studies (STREGA) Statement [23].

The buccal cell samples donated by the participants were collected in Resuspension Solution (GenElute Mammalian Genomic DNA Miniprep Kit, Sigma, St. Louis, MO, USA) with the use of sterile foam-tipped applicators (Puritan, LongIsland, NY, USA). DNA was extracted from the buccal cells using a GenElute Mammalian Genomic DNA Miniprep Kit (Sigma, St. Louis, MO, USA) according to the manufacturer’s protocol. All samples were genotyped in duplicate, using C__34384693_10 (for the *MMP1* rs1799750, -/G), C____632734_20 (for the *MMP10* rs486055, C/T) and C__15880589_10 (for the *MMP12* rs2276109, T/C) TaqMan^®^ Pre-Designed SNP Genotyping Assays (Applied Biosystems) on a StepOne Real-Time Polymerase Chain Reaction (RT-PCR) instrument (Applied Biosystems, Waltham, MA, USA) following the manufacturer’s recommendations. The PCR conditions were identical for all of the assays: 5 min of initial deneturation (95 °C), then 40 cycles of denaturation (15 s, 95 °C) and annealing/extension (60 s, 60 °C).

### Statistical Analyses

All statistical analyses were conducted in the R programming environment using specific R packages (version 3.4.0, The R Foundation for Statistical Computing, Vienna, Austria; https://cran.r-project.org.) Genotype and allele frequency distributions, in addition to HWE probabilities were determined under the different models of inheritance (codominant, dominant, recessive and over-dominant) using the *SNPassoc* package. Haplotype analysis was performed using the *haplo.stats* package. The haplo.score function was used to infer pseudo-haplotypes (a set of alleles located on the same chromosome) and to test the association between the generated allele combinations and the injury phenotype under the different models of inheritance (additive, dominant and recessive, respectively). The score statistic was used to assess the magnitude and direction of the haplotype-based association. *p* values < 0.05 were considered statistically significant.

## 3. Results

All genotype frequencies in the case and control groups conformed to the expectations of Hardy-Weinberg equilibrium (*p* = 0.582–1.0) as well as the pooled case-control sample (*p* = 0.562–0.616) (Table 1). Table 2 presents the genotype and allele frequency distributions for the three investigated SNPs within the *MMP1*, *MMP10* and *MMP12* genes. None of the polymorphisms were associated with the risk of non-contact ACL ruptures under the codominant, dominant, recessive and over-dominant models (sex adjusted). Likewise, no genotype–genotype interactions (sex adjusted) were observed to significantly associate with the risk of non-contact ACL ruptures (Table 3, Table 4 and Table 5). As all three genes are known to locate on the same chromosome, inferred haplotype analysis was also conducted. Eight haplotypes were observed which suggest that the variants were not tagged loci. Five common inferred haplotypes (frequencies > 5%) accounted for 97% of the observed variation. The most common haplotype was -/C/T (50.84%; rs1799750 - >G, rs486055 C > T, rs2276109 T > C, respectively). None of the inferred haplotypes were associated with non-contact ACL ruptures (sex adjusted) under the investigated genetic models (additive, dominant and recessive, respectively) (Table 6).

## 4. Discussion

The main findings of this genetic association study show (i) no independent associations between *MMP1* (rs1799750 - >G), *MMP10* (rs486055 C > T) and *MMP12* (rs276109 T > C) polymorphisms and non-contact ACL ruptures; (ii) no significant associations between any of the inferred haplotypes constructed from these variants with non-contact ACL ruptures; (iii) no significant gene–gene interactions between *MMP1*, *MMP10, MMP12* and non-contact ACL ruptures.

The 11q22 region encompasses a total of nine MMP genes, several of which have previously been investigated for their potential contribution to both acute (tendon and ligament ruptures) and chronic (tendinopathies) musculoskeletal soft tissue injuries. DNA variants located within *MMP1* (rs1799750), *MMP3* (rs3025058, rs679620, rs591058, rs650108), *MMP10* (rs486055), and *MMP12* (rs2276109), have independently and/or in combination been associated with Achilles tendinopathy (AT) risk [16,17,18], Carpal Tunnel Syndrome (CTS) [24] and ACL ruptures [19,20,21]^.^

Briefly, the original study by Raleigh et al. [16] reported the independent association of three *MMP3* variants (rs679620, rs591058 and rs650108) with the risk of AT. Furthermore, the same article described a significant allelic interaction between the functional *MMP3* rs679620 and *COL5A1* rs12722 variants, which was proposed to modify the risk of AT. El Khoury et al. [17] also described the association of the *MMP3* rs679620 and *TIMP2* rs4789932 variants with the risk of Achilles tendon injuries. Specifically, the *MMP3* rs679620 GG genotype was significantly overrepresented in the group of participants reporting acute Achilles tendon ruptures, whereas the *TIMP* rs4789932 CT genotype was associated with decreased risk when all males with Achilles tendon injuries were analysed). Further analysis by Gibbon et al. [18] described the significant association of the functional promoter variant, *MMP3* rs3205058, with the risk of AT in a South African cohort. In addition, an inferred haplotype constructed from four *MMP3* variants (rs3025058, rs679620, rs591058 and rs650108) was significantly associated with decreased risk for AT in an Australian cohort. However, these results contrasted to Raleigh et al. [16] and the authors cautioned against over-interpretation of the data, suggesting that the risk-conferring variants at the 11q22 chromosomal locus may still require identification.

Four DNA variants within *MMP1* (rs1799750), *MMP3* (rs679620), *MMP10* (rs486055) and *MMP12* (rs2276109) genes have also been suggested by Burger et al. [24] to be involved in the aetiology of idiopathic CTS. However, the authors observed no associations between any of the investigated *MMP* variants and CTS as well as no significant differences in the distributions of the inferred haplotypes constructed from the investigated MMP variants between the CTS and CON groups.

With respect to ACL injuries, Malila et al. [19] reported the significant overrepresentation of the of *MMP3* rs3025058 5A+ genotype in the group of participants reporting ACL ruptures sustained through contact mechanisms compared to the group of healthy controls. Subsequently, in a study by Posthumus et al. [20] four DNA variants mapping to the 11q22 chromosomal region (*MMP1* rs1799750, *MMP3* rs679620, *MMP10* rs486055, *MMP12* rs2276109) were investigated for predisposition to ACL ruptures in a South African cohort. This study reports the underrepresentation of the *MMP12* rs2276109 G allele in participants who sustained a non-contact mechanism of ACL ruptures compared to the controls. Furthermore, the authors reported that, although the *MMP3* rs679620 loci was not independently associated with ACL injury risk, the GG genotype trended towards statistical significance, more often presenting in the control group compared to the ACL rupture group. Interestingly, the *MMP3* rs679620 G allele was also consistently observed to be over-represented in the controls when the four-, three- and two-variant haplotypes were inferred. The authors suggested that the low-risk haplotype combinations may be associated with the presence of the G alleles for the *MMP3* rs679620 and *MMP12* rs2276109 variants [20].

In contrast with the results presented by Posthumus et al. [20], Lulińska-Kuklik et al. [21] demonstrated the significant underrepresentation of the *MMP3* rs679620 G allele in the control group compared to the group of participants reporting non-contact ACL injuries. Conversely, these results inferred the G allele to be associated with an increased risk for ACL injury. Further haplotype analysis identified the *MMP3* G-C (rs679620-rs591058) inferred haplotype as a potential risk- associated motif for non-contact ACL ruptures.

The current study also failed to demonstrate an association between the *MMP12* rs2276109 variant and ACL rupture risk, which was previously associated with ACL injuries in a South African cohort [20].

Despite the fact that the *MMP1* (rs1799750), *MMP10* (rs486055) and *MMP12* (rs2276109) genetic variants investigated in the present study were not associated with the risk of non-contact ACL ruptures, these genes remain important biological candidates for further interrogation based on the results from previous case-control genetic association studies, in addition to their functional contribution to matrix remodeling.

Matrix metalloproteinase-1 is ubiquitously expressed interstitial collagenase, which is involved in the breakdown of interstitial collagens types I, II, and III [25]. However, MMP1 overexpression is associated with several pathological musculoskeletal conditions, such as arthritis and tendinopathies [26,27,28]. In the study by Jones et al. [26], higher levels of MMP1 were detected in ruptured tendon samples compared to normal tendons. The *MMP1* rs1799750 deletion/insertion (-/G) polymorphism is located -1607 bp upstream from the transcription start site within the promoter region of the gene as is suggested to modulate transcriptional activity. In the presence of the *MMP1* rs1799750 G allele, the DNA sequence (5′-AAGAT-3′) is altered (5′-AAGGAT-3′), creating a binding site (5′-GGA-3) recognized by the members of the Ets family of transcription factors. This sequence change has been associated with increased MMP1 transcription, providing a plausible mechanism through which ECM degradation may be altered, thereby modulating injury susceptibility towards sustaining an acute musculoskeletal ruptures [25,26].

Stromelysin-2 (MMP10) degrades a variety of extracellular matrix proteins, including collagen types III–V, gelatin, casein, aggrecan, elastin and fibronectin. In addition, this stromelysin has the ability to activate other proteases, including MMP1, MMP7, MMP8, and MMP9 [9,29]. Interestingly, both painful and ruptured Achilles tendons have demonstrated distinct MMP10 mRNA profiles, suggesting differences in extracellular proteolytic activity [26]. The *MMP10* rs486055 (G/A) variant results in a non-synonymous substitution of an arginine (Arg) residue for a lysine (Lys) residue within the pro-peptide sequence. However, this polymorphism is not predicted to be deleterious using the SIFT (Sorting Intolerant From Tolerant) program. Furthermore, it is still to be determined whether this variant influences MMP10 mRNA expression. Therefore, it is evident that this variant needs to be further annotated before its possible contribution to the aetiology of musculoskeletal soft tissue injuries can be elucidated [20,24].

The macrophage metalloelastase (MMP12) is secreted by activated macrophages. This MMP demonstrates a broad substrate specificity responsible for the degradation of several ECM components, including laminin, fibronectin, vitronectin, type IV collagen, and heparan sulfate. A common SNP located in the *MMP12* gene promoter (rs2276109) results in the transition from an adenine to a guanine at the position -82. It was shown that this change can influence the binding of the activator protein-1 (AP-1) transcription factor in the promotor region of *MMP12*. Specifically, the A allele is associated with higher *MMP12* promoter activity through its ability to bind the AP-1 transcription factor with greater affinity [30]. Interestingly, in a study by Jones et al. [26] painful tendons demonstrated reduced levels of MMP12 expression compared to normal tendons, whereas ruptured samples demonstrated significantly elevated levels.

### Implications

Variants within the *MMP1*, (rs1799750 - >G), *MMP10* (rs486055 C > T) and *MMP12* (rs2276109 T > C) genes may not modulate susceptibility to ACL injuries.Interactions of different gene variants in several genes that code for structural and regulatory elements of ligaments are responsible for genetic susceptibility to ACL rupture.The interactions of MMP1 MMP10, MMP12 genetic variants in determining risk susceptibility may not contribute to future multifactorial risk models to ACL rupture.

## 5. Conclusions

Collectively, the published results have highlighted key differences in the expression profiles of specific MMPs between normal and diseased musculoskeletal soft tissues. A growing body of research proposes that DNA polymorphisms located within the MMP encoding genes may influence musculoskeletal phenotypes. However, this concept requires further elucidation before meaningful conclusions may be drawn. Despite advances in our understanding of the genetic basis of acute and chronic musculoskeletal soft tissue injuries, there are still limitations that have hampered the progression of genetic based research. Firstly, a potential limitation to this genetic association study is the sample size of the recruited ACLR participants, especially when the non-contact mechanism of injury is analyzed. Therefore, to overcome the obvious barrier, recruitment of larger group of ACLR patients with a non-contact mechanism of injury is needed. Secondly, as the current study did not strengthen evidence that variants located within the *MMP1, MMP10*, and *MMP12* gene influence non-contact ACL rupture risk, future research should aim to fully characterise the region encompassing all MMP genes mapping to chromosome 11. This level of fine mapping is now achievable with the advancement in high throughput sequencing technologies, chip arrays and associated bioinformatic analysis pipelines. Furthermore, new candidate variants identified using next-generation sequencing methods should be analysed in several independent cohorts to identify robust markers of risk.

## Figures and Tables

**Table 1 genes-11-00766-t001:** Minor allele frequencies (MAF) and Hardy-Weinberg equilibrium probabilities for the investigated variants (state variants) in the control and anterior cruciate ligament rupture groups investigated in the recruited Polish cohort.

SNP	MAF (%)	ACLR + CON	ACLR	CON
*MMP1* (*-/G* rs1799750)	allele G (47.7)	0.562	0.507	1.0
*MMP10* (C/T rs486055)	allele T (17.6)	0.616	0.461	0.826
*MMP12* (T/C rs2276109)	allele C (15.9)	0.591	1.0	0.582

MAF—minor allele frequency; ACLR—anterior cruciate ligament rupture, CON—control.

**Table 2 genes-11-00766-t002:** Association analysis of the *MMP1* rs1799750 -/G, *MMP10* rs486055 C/T and *MMP12* rs2276109 T/C polymorphisms with non-contact anterior cruciate ligament (ACL) rupture.

Polymorphism	Model	CON (*n* = 202)	%	ACLR ^#^ (*n* = 227/228)	%	OR	95% CI		*p* *
MMP1 rs1799750 -/G	Codominant								
	--	55	27.23	59	25.99	1.00			0.922/0.924
	-G	102	50.50	119	52.42	1.09	0.69	1.71	
	GG	45	22.28	49	21.59	1.02	0.59	1.75	
	Dominant								
	--	55	27.23	59	25.99	1.00			0.772/0.774
	-G + GG	147	72.77	168	74.01	1.07	0.69	1.64	
	Recessive								
	-- + -G	157	77.72	178	78.41	1.00			0.863/0.865
	GG	45	22.28	49	21.59	0.96	0.61	1.52	
	Overdominant								
	-- + GG	100	49.50	108	47.58	1.00			0.690/0.693
	-G	102	50.50	119	52.42	1.08	0.74	1.58	
MMP10 rs486055 C/T	Codominant								
	CC	131	64.85	163	71.49	1.00			0.335/0.400
	CT	63	31.19	58	25.44	0.74	0.48	1.13	
	TT	8	3.96	7	3.07	0.70	0.25	1.99	
	Dominant								
	CC	131	64.85	163	71.49	1.00			0.140/0.177
	CT + TT	71	35.15	65	28.51	0.74	0.49	1.11	
	Recessive								
	CC + CT	194	96.04	221	96.93	1.00			0.616/0.773
	TT	8	3.96	7	3.07	0.77	0.27	2.16	
	Overdominant								
	CC + TT	139	68.81	170	74.56	1.00			0.186/0.201
	CT	63	31.19	58	25.44	0.75	0.49	1.15	
*MMP12* rs2276109 T/C	Codominant								
	TT	144	71.29	158	69.30	1.00			0.678/0.584
	CT	55	27.23	64	28.07	1.06	0.69	0.69	
	CC	3	1.49	6	2.63	1.82	0.45	0.45	
	Dominant								
	TT	144	71.29	158	69.30	1.00			0.652/0.807
	CT + CC	58	28.71	70	30.70	1.10	0.73	0.73	
	Recessive								
	TT + CT	199	98.51	222	97.37	1.00			0.402/0.300
	CC	3	1.49	6	2.63	1.79	0.44	0.44	
	Overdominant								
	TT + CC	147	72.77	164	71.93	1.00			0.846/0.936
	CT	55	27.23	64	28.07	1.04	0.68	0.68	

ACLR—anterior cruciate ligament rupture, CON—control; * *p*-values/*p* value adjusted for sex; OR—odds ratio, 95% CI—confidence intervals, ^#^ 227 ACLR subjects for *MMP1* rs1799750 -/G, 228 ACLR subjects for *MMP10* rs486055 C/T and *MMP12* rs2276109 T/C.

**Table 3 genes-11-00766-t003:** Association analysis of the *MMP1 x MMP10* interaction with non-contact ACL rupture (codominant model).

*MMP1 x MMP10*		rs486055															*p **
		CC					CT					TT					
		CON (*n* = 131)	ACLR (*n* = 162)	OR	95% CI		CON (*n* = 62)	ACLR (*n* = 58)	OR	95% CI		CON (*n* = 8)	ACLR (*n* = 7)	OR	95% CI		
rs1799750	--	25	31	1.00			24	22	0.74	0.34	1.62	6	6	0.81	0.23	2.81	0.884/ 0.886
	-G	65	84	1.04	0.56	1.93	35	34	0.78	0.39	1.59	2	1	0.40	0.03	4.71	
	GG	41	47	0.92	0.47	1.81	4	2	0.40	0.07	2.38	0	0	NA	NA	NA	

ACLR—anterior cruciate ligament rupture group, CON—control group; * *p*-values/*p* value adjusted for sex; OR—odds ratio, 95% CI—confidence intervals; NA—not applicable.

**Table 4 genes-11-00766-t004:** Association analysis of the *MMP1 x MMP12* interaction with non-contact ACL rupture (codominant model).

*MMP1 x MMP12*		rs2276109															*p **
		TT					CT					CC					
		CON (*n* = 144)	ACLR (*n* = 158)	OR	95% CI		CON (*n* = 55)	ACLR (*n* = 63)	OR	95% CI		CON (*n* = 3)	ACLR (*n* = 6)	OR	95% CI		
rs1799750	--	48	51	1.00	NA	NA	6	8	1.25	0.41	3.88	1	0	0.00	0.00	NA	0.412/ 0.347
	-G	71	84	1.11	0.67	1.85	31	35	1.06	0.57	1.98	0	0	NA	NA	NA	
	GG	25	23	0.87	0.43	1.73	18	20	1.05	0.49	2.21	2	6	2.82	0.54	14.67	

ACLR—anterior cruciate ligament rupture, CON—control; * *p*-values/*p* value adjusted for sex; OR—odds ratio, 95% CI—confidence intervals; NA—not applicable.

**Table 5 genes-11-00766-t005:** Association analysis of the *MMP10 x MMP12* interaction with non-contact ACL rupture (codominant model).

*MMP10 x MMP12*		rs2276109															*p **
		TT					TC					CC					
		CON (*n* = 144)	ACLR (*n* = 158)	OR	95% CI		CON (*n* = 55)	ACLR (n = 64)	OR	95% CI		CON (*n* = 3)	ACLR (*n* = 6)	OR	95% CI		
rs486055	CC	91	111	1.00	NA	NA	38	47	1.01	0.61	1.69	2	5	2.05	0.39	10.81	0.586/ 0.553
	CT	46	43	0.77	0.46	1.26	16	14	0.72	0.33	1.55	1	1	0.82	0.05	13.29	
	TT	7	4	0.47	0.13	1.65	1	3	2.46	0.25	24.05	0	0	NA	NA	NA	

ACLR—anterior cruciate ligament rupture, CON—control; * *p*-values/*p* value adjusted for sex; OR—odds ratio, 95% CI—confidence intervals; NA—not applicable.

**Table 6 genes-11-00766-t006:** Haplotype-based association of *MMP1*, *MMP10* and *MMP12* polymorphisms (rs1799750, rs486055, rs2276109) with non-contact ACL rupture.

Haplotype [rs1799750, rs486055, rs2276109]	Frequency (%)			Additive (Score = 2.95, *p* = 0.890)		Dominant (Score = 2.40, *p* = 0.966)		Recessive (Score = 2.30, *p* = 0.680)	
	ACLR + CON	ACLR	CON	Score	*p* *	Score	*p* *	Score	*p* *
[GTT]	8.55	7.24	10.37	-1.29	0.197/0.225	-1.22	0.221/0.239	NA	NA
[-TT]	6.87	6.23	7.61	-0.71	0.480/0.613	-0.49	0.626/0.648	-0.89	0.372/0.441
[-CC]	0.60	0.53	0.68	-0.21	0.836/0.648	-0.21	0.836/0.769	NA	NA
[-TC]	1.27	1.23	1.30	-0.16	0.870/0.795	-0.16	0.870/0.795	NA	NA
[-CT]	17.81	17.95	17.64	0.08	0.936/0.979	-0.06	0.949/0.917	0.25	0.802/0.850
[GTC]	0.87	1.09	0.28	0.16	0.870/0.811	0.16	0.870/0.887	NA	NA
[GCT]	50.84	51.91	49.29	0.48	0.634/0.638	0.22	0.824/0.811	0.60	0.549/0.503
[GCC]	13.20	13.81	12.84	0.77	0.443/0.487	0.55	0.580/677	0.98	0.325/0.240

ACLR—anterior cruciate ligament rupture, CON—control; NA—not applicable, * *p*-values/*p* value adjusted for sex.

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
