# Peer review of "Matrix Metalloproteinase Genes (MMP1, MMP10, MMP12) on Chromosome 11q22 and the Risk of Non-Contact Anterior Cruciate Ligament Ruptures"

_genes, 2020, doi:10.3390/genes11070766_

Round 1

Reviewer 1 Report

Reviewer’s comments:

The manuscript with the ID: genes-846899 entitled: “Matrix metalloproteinase genes (MMP1, MMP10, 2 MMP12) on chromosome 11q22 and risk of non-contact anterior cruciate ligament ruptures”, aims to explore the possible role of three SNPs in three genes localized at chromosome 11 and encoding MMPs as possible risk factors of  anterior cruciate ligament (ACL) rupture in a Polish cohort.

These SNPs are the following MMP1 (rs1799750, -27 >G), MMP10 (rs486055, C>T) and MMP12 (rs2276109, T>C)

Although the main findings did not provide evidence for the influence of these SNPs on the non- contact ACL rupture risk, the manuscript is interesting and will help to the scientists searching for the candidate-genes and polymorphisms for the risk of anterior cruciate ligament rupture.

The study is well designed, carried out and presented. The study populations are quite small (228 with ligament rupture. and 202 controls) but still sufficient in size to get reliable results. The description of the individuals with non-contact anterior cruciate ligament rupture and the findings are presented clearly with required thoroughness in 6 tables. The methods are appropriate and reliable for obtaining trustworthy results. The statistical analyses are also appropriate and adequate for realization of the intended tasks.  

The introduction, discussion and conclusions are adequately supported by literature overview, and obtained data. The title and the abstract convey the results described in the manuscript.

I consider that the manuscript will be improved after some minor corrections.

Minor comments

  1. Introduction: line 57 – elastin IS NOT a glycoprotein, that’s why elastin should excluded from the brackets. Elastin is a protein associated with glycoproteins (e.g. fibrillin) in the elastic fibers.
  2. Materials and methods: Genotyping: (lines 125-129) - The thermal profile of the PCR reactions should be added.
  3. Results: In addition to the presented data of the association analysis (Table 2), I would recommend that the same type of associations could be also performed in the subgroups of males and females.
  4. Discussion: line 13: before the AT as abbreviation the whole text meaning should be given (I guess, Achilles tendinopathy, AT)
  5. Conclusions: lines 105-106: “Collectively, these results highlight key differences in the expression profiles of specific MMPs between normal and diseased musculoskeletal soft tissues.” – Which results do the authors mean? If they consider their own results – the obtained results in the manuscript do not support the notion. If the authors means the discussed results of other research groups – they this should be given clearly.

Author Response

The authors would like to thank the reviewers for the comments. We have tried to address the comments in in BOLD type set:

Reviwer 1 #

Reviewer’s comments:

The manuscript with the ID: genes-846899 entitled: “Matrix metalloproteinase genes (MMP1, MMP10, 2 MMP12) on chromosome 11q22 and risk of non-contact anterior cruciate ligament ruptures”, aims to explore the possible role of three SNPs in three genes localized at chromosome 11 and encoding MMPs as possible risk factors of  anterior cruciate ligament (ACL) rupture in a Polish cohort.

These SNPs are the following MMP1 (rs1799750, -27 >G), MMP10 (rs486055, C>T) and MMP12 (rs2276109, T>C)

Although the main findings did not provide evidence for the influence of these SNPs on the non- contact ACL rupture risk, the manuscript is interesting and will help to the scientists searching for the candidate-genes and polymorphisms for the risk of anterior cruciate ligament rupture.

The study is well designed, carried out and presented. The study populations are quite small (228 with ligament rupture. and 202 controls) but still sufficient in size to get reliable results. The description of the individuals with non-contact anterior cruciate ligament rupture and the findings are presented clearly with required thoroughness in 6 tables. The methods are appropriate and reliable for obtaining trustworthy results. The statistical analyses are also appropriate and adequate for realization of the intended tasks. 

The introduction, discussion and conclusions are adequately supported by literature overview, and obtained data. The title and the abstract convey the results described in the manuscript.

I consider that the manuscript will be improved after some minor corrections.

Minor comments

Introduction: line 57 – elastin IS NOT a glycoprotein, that’s why elastin should excluded from the brackets. Elastin is a protein associated with glycoproteins (e.g. fibrillin) in the elastic fibers.

Reply –The sentence in line 55 was edited to read: “ The fibrous connective tissue of tendons and ligaments is composed of numerous collagenous fibre types (e.g. I, III-VI and, XII) and minor elastic fibres (e.g. elastin) as well as other non-collagenous particles such as proteoglycans (e.g. decorin, lumican and aggrecan) and glycoproteins (e.g. tenascin C) [7].

Materials and methods: Genotyping: (lines 125-129) - The thermal profile of the PCR reactions should be added.

Reply –The thermal profile of the PCR reactions have been added to the manuscript

Results: In addition to the presented data of the association analysis (Table 2), I would recommend that the same type of associations could be also performed in the subgroups of males and females.

Reply -  

In table 2, all analyses were already adjusted for participants’ sex (used as an additive term in all model, because models with the interaction terms  (genotype x sex) were not significantly better than models without the interaction term( LRT, Likelihood Ratio Test):

MMP1_rs1799750: codominant, p=0.864, dominant, p=0.603, recessive, p=0.982, overdominant, p=0.659

MMP10_rs486055: codominant, p=0.121,  dominant, p=0.202, recessive, p=0.053, overdominant, p=0.640

MMP12_rs2276109: codominant, p=0.997, dominant, p=0.938, recessive, p=0.957, overdominant, p=0.928

Thus, subgroups analysis, which would also be of less power than the whole group analysis, was not necessary.

In tables 3-5, all analyses were also adjusted for sex (used as an additive term, because gene-gene interaction models with the interaction term for sex were not significantly better than gene-gene interaction model without the interaction with sex (LRT:

MMP1 x MMP10: p=0.126

MMP1 x MMP12: p=0.997

MMP10 x MMP12: p=0.995

Discussion: line 13: before the AT as abbreviation the whole text meaning should be given (I guess, Achilles tendinopathy, AT)

Reply –The following was added in line 13 to address the matter: “Achilles tendinopathy (AT) risk”

Conclusions: lines 105-106: “Collectively, these results highlight key differences in the expression profiles of specific MMPs between normal and diseased musculoskeletal soft tissues.” – Which results do the authors mean? If they consider their own results – the obtained results in the manuscript do not support the notion. If the authors means the discussed results of other research groups – they this should be given clearly.

Reply – The authors are referring to the collective published data and therefore the sentence in line 105-106 was adjusted to read as follows: “Collectively, the published results have highlighted key differences in the expression profiles of specific MMPs between normal and diseased musculoskeletal soft tissues.”

Reviewer 2 Report

Ms. No.: genes-846899

Title: Matrix metalloproteinase genes (MMP1, MMP10, MMP12) on chromosome 11q22 and risk of non-contact anterior cruciate ligament ruptures

Reviewer comments

The focus of this manuscript is a study of a “11q22 region,” which “encompasses a total of nine MMP genes.” The authors tested whether there exists an association between genes regulating the expression of matrix metalloproteinases (MMPs) 1, 10, and 12 and non-contact anterior cruciate ligament (ACL) ruptures. This hypothesis is based on a similar correlation between MMP-3 and ACL, which is (according to authors) well-documented in the literature on the subject. Authors enrolled 430 participants of both sexes (age: below 30 years), physically active (9 hrs per week on average), semi- and professional football (US: soccer) players. This extensive study revealed no association between tested variables.

Considering authors are in possession of all the demographic information, one wonders if splitting respective test groups into different or smaller subsets wouldn’t reveal any correlations which are obscured by pooling all the samples (CON vs. ACL). For example, does the sex of the participants has any effect? Or participation in the top tier versus lover divisions?   

A minor technical issue with formatting: line numbering is no longer consecutive after page 7. Possibly the tables are an issue.

Specific comments

Line 40: “Despite the fact the current study did not support existing evidence suggesting that variants within the MMP1, MMP10, and MMP12 genes influence non-contact ACL rupture risk, future work should include high-throughput sequencing technologies to identify potential targeted polymorphisms to fully characterize the 11q22 region with susceptibility to non-contact ACL rupture susceptibility in a Polish cohort.”

These conclusions look like a recommendation for future studies. What conclusions can be drawn from the experimental data analyzed within the scope of performed work on an enrolled cohort? What was the hypothesis tested? Did you find evidence that supports the hypothesis?

Line 186: “Table 2. Association analysis of the…” Please, report the results by using only two significant digits. It also applies to the results presented in the other tables, particularly table 6.

Page 8, line 1: “The main findings of this genetic association study that aimed to investigate the possible 2 interaction between three MMP genes variants spanning the chromosome 11q22 region and non-3 contact ACL ruptures in a Polish cohort were:”

This sentence should be re-written, considering it contains the introduction and simultaneously reports the findings. Both are intertwined in a typical Polish style of writing, which doesn’t help readability. Consequently, a single sentence is 7 lines long (!).

Author Response

The authors would like to thank the reviewers for the comments. We have tried to address the comments in in BOLD type set:

Reviewer comments

The focus of this manuscript is a study of a “11q22 region,” which “encompasses a total of nine MMP genes.” The authors tested whether there exists an association between genes regulating the expression of matrix metalloproteinases (MMPs) 1, 10, and 12 and non-contact anterior cruciate ligament (ACL) ruptures. This hypothesis is based on a similar correlation between MMP-3 and ACL, which is (according to authors) well-documented in the literature on the subject. Authors enrolled 430 participants of both sexes (age: below 30 years), physically active (9 hrs per week on average), semi- and professional football (US: soccer) players. This extensive study revealed no association between tested variables.

Considering authors are in possession of all the demographic information, one wonders if splitting respective test groups into different or smaller subsets wouldn’t reveal any correlations which are obscured by pooling all the samples (CON vs. ACL). For example, does the sex of the participants has any effect? Or participation in the top tier versus lover divisions?  

Reply – Subgroup analyses were not conducted, as there were no significant genotype x sex interaction. The association analyses in table 2 and 3-5 were sex adjusted (sex as an additive term).

A minor technical issue with formatting: line numbering is no longer consecutive after page 7. Possibly the tables are an issue.

Reply -  Line numbering has been fixed

Specific comments

Line 40: “Despite the fact the current study did not support existing evidence suggesting that variants within the MMP1, MMP10, and MMP12 genes influence non-contact ACL rupture risk, future work should include high-throughput sequencing technologies to identify potential targeted polymorphisms to fully characterize the 11q22 region with susceptibility to non-contact ACL rupture susceptibility in a Polish cohort.”

These conclusions look like a recommendation for future studies. What conclusions can be drawn from the experimental data analyzed within the scope of performed work on an enrolled cohort? What was the hypothesis tested? Did you find evidence that supports the hypothesis?

Reply – We acknowledge that our research had not identified any associations with any of the polymorphisms within the 11q22 region to susceptibility to non-contact ACL rupture in a Polish cohort. Given the variability in the associations at this locus and taken together with the functional evidence of these proteins and their altered expression profiles noted in pathological tissue this locus remains suitable for further analyses. We had only tested discreet polymorphisms and there exists a potential that other variants in LD with the true risk conferring clinically relevant polymorphism can be explored in this region. Technologies such as sequencing would need to be applied. For these reasons, the authors believe that this statement made in the discussion holds relevance.

Line 186: “Table 2. Association analysis of the…” Please, report the results by using only two significant digits. It also applies to the results presented in the other tables, particularly table 6.

Reply – change has been made

Page 8, line 1: “The main findings of this genetic association study that aimed to investigate the possible 2 interaction between three MMP genes variants spanning the chromosome 11q22 region and non-3 contact ACL ruptures in a Polish cohort were:”

This sentence should be re-written, considering it contains the introduction and simultaneously reports the findings. Both are intertwined in a typical Polish style of writing, which doesn’t help readability. Consequently, a single sentence is 7 lines long.

Reply -  The sentence on Page 8, Line 1 was adjusted to read as follows: “The main findings of this genetic association study showed i) no independent associations between MMP1 (rs1799750 ->G), MMP10 (rs486055 C>T) and MMP12 (rs276109 T>C) polymorphisms and non-contact ACL ruptures ii) no significant associations between any of the inferred haplotypes constructed from these variants with non-contact ACL ruptures; iii) no significant gene-gene interactions between MMP1, MMP10, MMP12 and non-contact ACL ruptures.”